# Reversible On/Off Switching of Ferroelectricity in a Molecular FeCo Prussian Blue Analogue with Multiple Control

Yu-Bo Huang[1,2], Sheng-Qun Su[1,3] ✉, Wen-Huang Xu[1,4], Wen-Wei Zheng[1], Tian-Chi Ji[1,5], Kai-Ge Gao[6], Hui-Hui Cui[7], Shimon Ikenaga[8], Kaoru Yamamoto [9], Zhao-Yang Li [10], Shu-Qi Wu [1] ✉ & Osamu Sato [1] ✉

Polar molecule-based magnetic materials capable of multistate switching have garnered significant interest for their potential applications in next-generation memory devices, sensors, and energy conversion. Among such materials, Prussian blue analogs that exhibit electron transfer–coupled spin transition (ETCST) behavior stand out due to their unique switching properties. In this study, we report a trinuclear cyanide-bridged FeCo compound (**1**) that exhibits photo- and thermo-induced ferroelectric phase transition via ETCST mechanism. Notably, this FeCo complex also demonstrates a quenching effect, whereby a non-polar state is trapped as a metastable state at low temperature, allowing the polarization control via the cooling rate variations. In addition, reversible single-crystal-to-single-crystal transformation via the desorption and absorption of solvent molecules are observed. The EtOH-removed FeCo compound (**1′**) loses its ferroelectricity, revealing that the ferroelectric phase transition can be modulated by dynamic desorption/adsorption of the guest molecule. These findings advance the design of functional ferroelectric materials with multistate switching capabilities, offering potential for future applications in contactless memory devices and beyond.

Ferroelectric memory devices are distinguished by their inherent ability to retain data in the absence of a continuous power supply, thus offering substantial energy-saving advantages[1–3]. Conventional ferroelectric materials typically modulate macroscopic polarization through mechanisms such as order–disorder transitions, ion displacements, and intermolecular charge transfer[4–8]. However, these systems often encounter integration challenges stemming from the unavoidable physical contact between electrodes and the ferroelectric layer, underscoring the need for novel strategies to control ferroelectricity[9,10]. In addition, the remnant polarization of ferroelectric materials tends to diminish with repeated switching cycles (fatigue), necessitating effective anti-fatigue solutions[11–14]. One promising approach is to exploit guest molecules to toggle ferroelectricity on and off, enabling the reset of ferroelectric states without resorting to conventional thermal cycling.

[1]Institute for Materials Chemistry and Engineering and IRCCS, Kyushu University, 744 Motooka, Nishi-ku, Fukuoka, Japan. [2]Institute for Solid State Physics, The University of Tokyo, Kashiwa, Chiba 277-8581, Japan. [3]Department of Chemistry, School of Science, Institute of Science Tokyo, 2-12-1, O-okayama, Meguro-ku, Tokyo, Japan. [4]State Key Laboratory of Optical Fiber and Cable Manufacture Technology, Yangtze Optical Fibre and Cable joint stock limited company; Optics Valley Laboratory, Hubei, PR China. [5]Shenyang National Laboratory for Materials Science, Institute of Metal Research, Chinese Academy of Sciences, 72 Wenhua Road, Shenyang, Liaoning, PR China. [6]College of Physical Science and Technology, Yangzhou University, Jiangsu, PR China. [7]School of Chemistry and Chemical Engineering, Nantong University, Nantong, PR China. [8]Graduate School of Science and Engineering, Okayama University of science, 1-1 Ridaicho, Okayama, Japan. [9]Department of Physics, Faculty of Science, Okayama University of Science, 1-1 Ridaicho, Okayama, Japan. [10]School of Materials Science and Engineering, Nankai University, Tianjin, PR China. ✉e-mail: shengquns@gmail.com; wu.shuqi.152@m.kyushu-u.ac.jp; sato@cm.kyushu-u.ac.jp

Dynamic molecular materials capable of adopting multiple states have attracted growing interest for their ability to switch various physical properties–such as magnetism and electric polarization–under external stimuli[15–23]. To date, most of these systems have been primarily studied for magnetic switching, driven by mechanisms such as spin crossover, electron transfer, and structural transformations[20,22,24–29]. Although many of these studies have focused on magnetic switching mechanisms (e.g., spin crossover, electron transfer, and structural transformations), recent findings in molecular Prussian blue analogs demonstrate that thermal- and visible-light-induced polarization switching can be realized through electron transfer-coupled spin transition (ETCST)[30,31]. Here, directional intramolecular electron transfer induces pronounced changes in molecular dipole moments, thereby enabling macroscopic polarization switching and paving the way for contactless, light-driven memory devices. Despite this progress, there still lacks the direct evidence of electric-field control of polarization in these systems, and the full range of strategies for manipulating polarization remains far from exhausted.

The particularly compelling feature of molecular Prussian blue analogs lies in their long-lived electron-transferred metastable states, which can be accessed with high efficiency via non-thermal channels such as light irradiation or rapid cooling[32]. This capability has led to the observation of multistep phase transitions and hidden phases with distinct properties[33–40]. Furthermore, solvent molecules could provide an additional freedom of tunability in cyanide-bridged compounds, exemplified by molecular magnetic sponges during guest absorption–desorption processes[41–43]. Given the significant dipole changes associated with ETCST, integrating it with ferroelectricity facilitates access to a rich variety of polarization states (vanishing, positive and negative), each responsive to certain stimuli. As a result, multistate polarization manipulation can be achieved by application of electric fields, photoinduced, thermally induced, and guest molecule–dependent processes (Fig. 1). In particular, the reversible

generation and erasure of ferroelectricity via metastable states created through light exposure or thermal quenching opens up new opportunities for circumventing fatigue, as guest molecule–driven on/off switching of ferroelectricity can be used to reset the material through dynamic desorption–absorption cycles.

In this study, we developed a compound exhibiting directional ETCST by incorporating distinct solvent molecules based on our previous work[30], i.e., [Fe(Tp)(CN)$_3$]$_2$[Co(dpa)$_2$]·2H$_2$O·1.5EtOH (1) (Tp = trispyrazolylborate, dpa = 2,2'-dipyridylamine, EtOH = ethanol) to achieve multistate polarization switching (Fig. S1a). As anticipated, 1 undergoes a nonpolar-to-polar phase transition during the ETCST process, a distinguishing feature among Prussian blue analogs. Its ferroelectric nature is confirmed by the reversal of spontaneous polarization under an applied electric field. Notably, the metastable Fe$^{III}_{LS}$(μ-CN)Co$^{II}_{HS}$ state can be trapped as an MS* (quenched) state at low temperature via thermal quenching (rapid cooling), which closely resembles its photoinduced metastable phase (MS) and high-temperature (HT) phase. Moreover, a controlled thermal treatment yields the ethanol-removed compound [Fe(Tp)(CN)$_3$]$_2$[Co(dpa)$_2$]·2H$_2$O (1') via a single-crystal-to-single-crystal transition, which undergoes a distinct ETCST process without polarization switching–highlighting the crucial role of solvent molecules in modulating ferroelectricity. These trapping effects, in combination with tunable vapor absorption–desorption, enable the formation of multiple states, including the HT, low-temperature (LT), quenched, and photoinduced MS in 1, as well as the HT and LT states in 1'. These states were characterized by single-crystal X-ray diffraction (XRD), infrared (IR) spectroscopy, and magnetic susceptibility measurements, and polarization switching behavior could be directly tracked by pyroelectric measurements. Therefore, in addition to conventional light- and thermal-driven ferroelectric phase transitions, we demonstrate polarization control through both the cooling rate and the dynamic desorption–adsorption of guest molecules.

## Polarization & Electron Transfer & Spin Transition

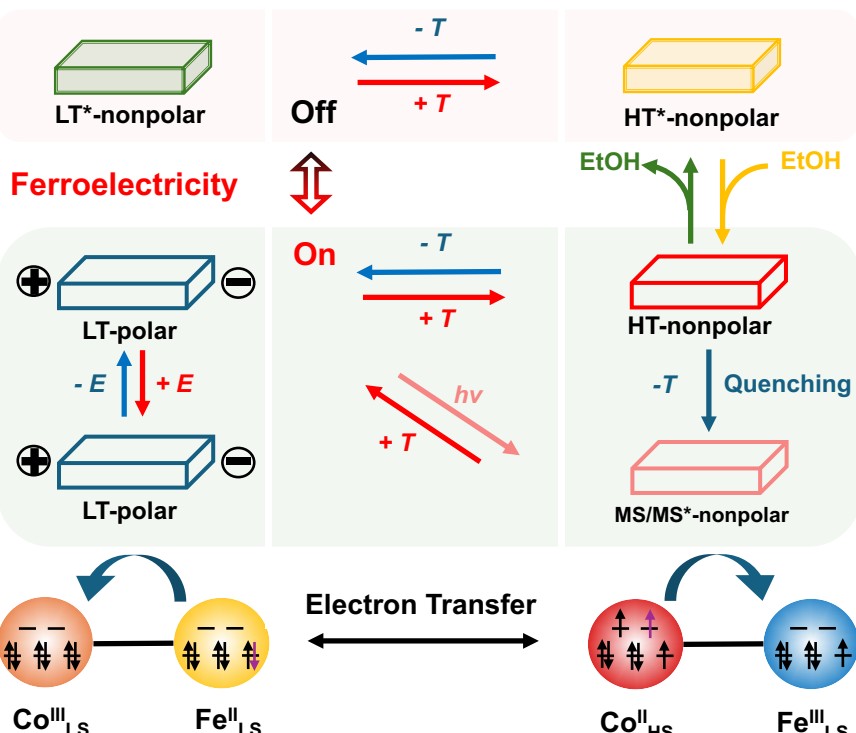

**Fig. 1 | Multistate control of polarization switching via photo- and thermal-induced, and guest molecule–dependent ETCST processes.** HT high-temperature phase; LT low-temperature phase; MS metastable state; HT* high-temperature phase after desorption; LT* low-temperature phase after desorption; T temperature; E electric field.

## Results

The temperature-dependent direct-current magnetic susceptibility of **1** was measured with an external magnetic field of 2000 Oe at a temperature range of 5–200 K using freshly filtered microcrystalline samples (Figs. 2, S1b). Below 140 K, the product of magnetic susceptibility and temperature ($\chi_m T$) was maintained at *ca.* 0.78 cm³ mol⁻¹ K, which was consistent with an Fe$^{III}_{LS}$ ion with $S = 1/2$, suggesting that the diamagnetic pair [Fe$^{II}_{LS}$–Co$^{III}_{LS}$] was formed in the LT phase. Further heating caused an abrupt increase in the $\chi_m T$ to *ca.* 4.05 cm³ mol⁻¹ K

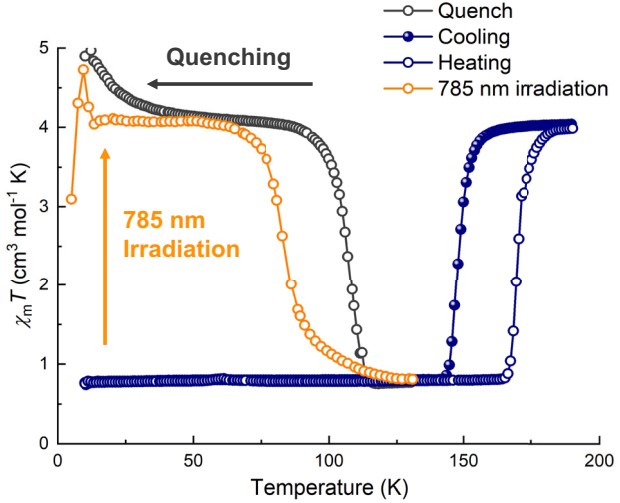

**Fig. 2 | Magnetic susceptibility of 1. Temperature-dependent $\chi_m T$ product of thermal-, 785 nm light- and quenching effect-induced ETCST behavior.** The measurements prior to and after irradiation were performed at a sweeping rate of 2 K min⁻¹. Open circle, heating process; filled circle, cooling process.

above 184 K, which was higher than the spin-only value obtained with an uncoupled Co$^{II}_{HS}$ ion ($S = 3/2$) and two Fe$^{III}_{LS}$ ions ($S = 1/2$) in the HT phase. This is caused by the significant unquenched orbital angular momentum for both ions in octahedral ligand fields. During the cooling process, the same transition of magnetic susceptibility occurred at a different temperature. By analyzing the $\chi_m T$ with respect to temperature, phase transition temperatures of $T_{1/2}(\downarrow) = 148$ K and $T_{1/2}(\uparrow) = 170$ K with a hysteresis of 22 K were obtained. Thus, the reversible bistability from the thermal-induced ETCST between the LT phase [Fe$^{II}_{LS}$–Co$^{III}_{LS}$–Fe$^{III}_{LS}$] and HT phase [Fe$^{III}_{LS}$–Co$^{II}_{HS}$–Fe$^{III}_{LS}$] was confirmed. This process was strongly supported by recording variable-temperature IR spectra from 140 to 200 K (Fig. S2). In the LT phase, vibration bands observed at 2089 and 2187 cm⁻¹ were attributable to the bridging $\nu_{CN}$ stretching mode of the Fe$^{II}_{LS}$(μ-CN)Co$^{III}_{LS}$ and Fe$^{III}_{LS}$(μ-CN)Co$^{III}_{LS}$ linkages, respectively. Two absorptions at 2053 and 2062 cm⁻¹ were attributable to the terminal cyanides of [Fe$^{II}$(Tp)(CN)₃]²⁻, whereas only one peak at 2131 cm⁻¹ was ascribable to the $\nu_{CN}$ mode of the uncoordinated cyanide of [Fe$^{III}$(Tp)(CN)₃]⁻. As the temperature increased above 165 K, the peak at 2187 cm⁻¹ gradually declined and new bands were observed at 2127 and 2164 cm⁻¹, assigned to the terminal $\nu_{CN}$ absorption of [Fe$^{III}$(Tp)(CN)₃]⁻ and the bridging $\nu_{CN}$ absorption of Fe$^{III}_{LS}$(μ-CN)Co$^{II}_{HS}$, respectively[40,44–46]. In addition, the ultraviolet–visible–near IR (UV–VIS–NIR) absorption spectra of **1** were recorded in the HT and LT phases (Fig. S3). The UV–VIS–NIR spectrum of the LT phase exhibited a broad absorption peak centered at 880 nm, which was assignable to the Fe$^{II}$ → Co$^{III}$ intervalence charge transfer band[40,44–46]. With an increase in temperature, the absorption of this peak disappeared in the HT phase.

Single-crystal structures of freshly filtered **1** were collected to characterize the structural rearrangements during thermal-induced ETCST processes (Fig. 3). Notably, in the LT phase, **1** was observed to adopt a polar crystal structure in the *Pn* space group (Table S1). The asymmetric unit comprised two V-shaped neutral [Fe(Tp)(CN)₃]₂[Co(dpa)₂] motifs, four water molecules, and three EtOH

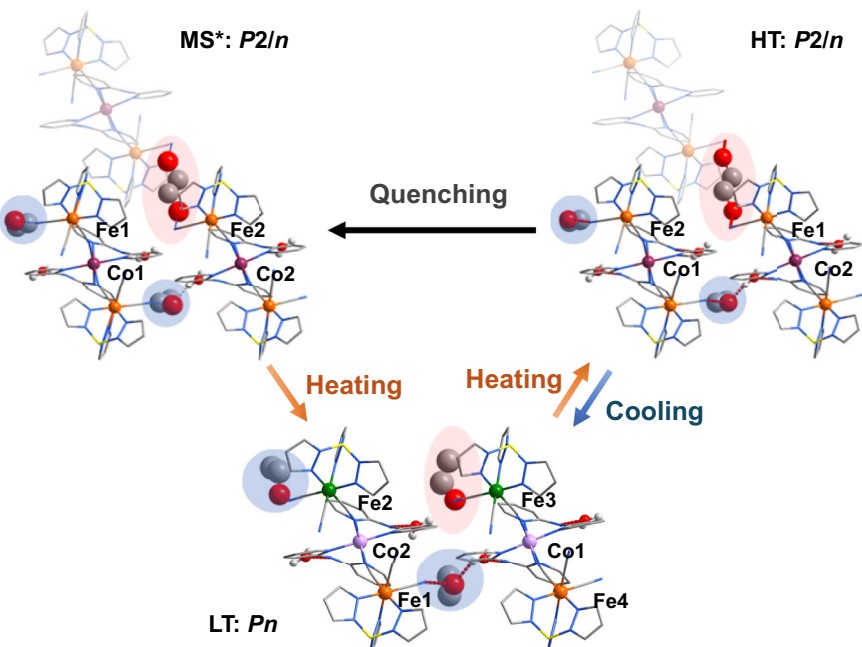

**Fig. 3 | Molecular structure and space group switching of 1.** The LT phase was in the polar space group *Pn*, whereas the HT and MS* phases were in the nonpolar space group *P2/n*. The MS* phase was obtained by placing the single-crystal sample directly at 30 K, and no ETCST behavior occurred. With a temperature increase, the MS* phase returned to the LT phase via relaxation. During the thermal-induced ETCST process from the HT to LT phases, the orientation of part of the EtOH

molecules changed from disorder to order. The gray dashed lines represent hydrogen bonds. The faintly drawn part is the adjacent structure with a disordered EtOH molecule connected by hydrogen bonds. Ethanol molecules are indicated with an ellipse (type A, red; type B, blue). Orange, Fe$^{III}$; green, Fe$^{II}$; purple, Co$^{II}$; pink, Co$^{III}$; yellow, B; gray, C; red, O; blue, N; light gray, H.

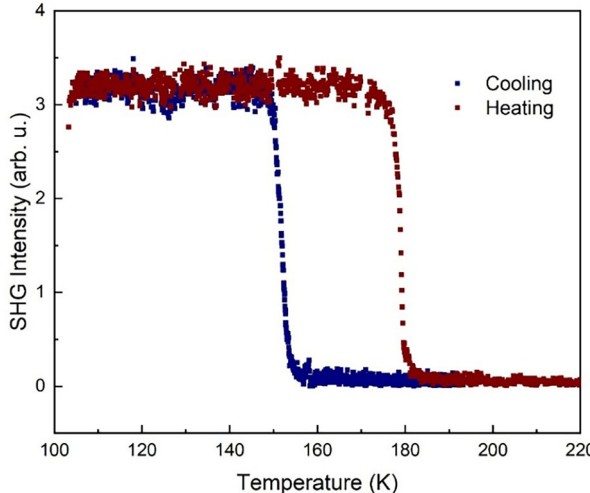

**Fig. 4 | Temperature dependence of the SHG intensity of 1.** The emergence of SHG at low temperature indicates a hysteretic, symmetry-breaking phase transition. Blue squares: cooling; red squares: heating.

molecules. The Co–N bond lengths ranged from 1.893(7) to 1.954(7) Å, suggesting the typical coordination environment of a $Co^{III}_{LS}$ ion. The Fe ions were coordinated by three N atoms of a Tp ligand and three C atoms from cyanides, resulting in two pairs of chemically nonequivalent Fe ions (Fe1, Fe2 and Fe3, Fe4), which were characterized by the first coordination sphere as follows. The bond lengths of Fe1–N and Fe1–C were in ranges of 1.998(6)–2.007(6) Å and 1.850(7)–1.911(8) Å, respectively. The Fe2–N and Fe2–C bond lengths were in ranges of 1.956(6)–1.993(7) Å and 1.905(8)–1.937(8) Å, respectively. The Fe3–N and Fe3–C bond lengths ranged from 1.961(6) to 1.980(6) Å and 1.908(7) to 1.941(7) Å, respectively. Finally, the Fe4–N and Fe4–C bond lengths ranged from 2.003(6) to 2.009(6) Å and 1.869(7) to 1.911(8) Å, respectively (Table S2). The differences in the bond lengths of Fe–N and Fe–C between both pairs of Fe ions, which exceeded the experimental error, represented the difference between $Fe^{III}_{LS}$ and $Fe^{II}_{LS}$, compared with the reported structures constructed with Tp building blocks. This indicated that Fe2, Fe3 corresponded to $Fe^{III}_{LS}$, and Fe1, Fe4 corresponded to $Fe^{II}_{LS}$, respectively[30,47–50]. The bond length data indicated that the formation of [$Fe^{II}_{LS}$–$Co^{III}_{LS}$–$Fe^{III}_{LS}$], $Fe^{II}$, and $Fe^{III}$ was distinguishable. In the crystal lattice, one ordered ethanol molecule forms a hydrogen bond with a non-bridging cyanide group, while each of the other two ethanol molecules simultaneously engages in hydrogen bonding with both a water molecule and another non-bridging cyanide group. All water molecules participate in hydrogen bonding with the nitrogen atoms of the dpa ligands. Through these interactions, the trinuclear [Fe₂Co] units are further extended into a three-dimensional network stabilized by hydrogen bonds involving both water and ethanol molecules.

As the temperature increased to the HT phase, the space group changed to *P*2/*n* with a nonpolar structure. The asymmetric unit comprised two half [Fe(Tp)(CN)₃]₂[Co(dpa)₂] trinuclear motifs, two water molecules, one ordered EtOH molecule, and a half disordered EtOH molecule. The Co–N bond lengths of both Co ions ranged from 2.119(2) to 2.167(2) Å, suggesting the typical coordination of $Co^{II}_{HS}$. The Fe–N and Fe–C bond lengths of both Fe ions ranged from 1.968(2) to 1.986(2) Å and 1.909(2) to 1.935(2) Å, respectively, which were common values for $Fe^{III}_{LS}$ ions. The bond length data indicated the formation of [$Fe^{III}_{LS}$–$Co^{II}_{HS}$–$Fe^{III}_{LS}$]. Interestingly, one of the EtOH molecules, which forms hydrogen bonds with the uncoordinated cyanide group of $Fe^{II}$ in the LT phase, was disordered in the HT phase because of the oxidation of $Fe^{II}$. These features of the HT and LT phases were consistent with that of a thermal-induced directional ETCST

between the [$Fe^{II}_{LS}$–$Co^{III}_{LS}$–$Fe^{III}_{LS}$] and [$Fe^{II}_{LS}$–$Co^{II}_{HS}$–$Fe^{III}_{LS}$] phases, implying a thermal-induced ferroelectric phase transition process. The remaining ethanol and water molecules exhibit only minor changes in orientation. From the perspective of crystal packing, the directional electron transfer process is accompanied by an order–disorder transition of the ethanol molecules. This observation highlights the crucial role of ethanol in facilitating the ferroelectric phase transition. These findings underscore the importance of guest molecular dynamics in governing the switchable ferroelectric properties of the system.

Moreover, the optical second-harmonic generation (SHG) measurements on crushed single crystal samples were also performed to investigate the symmetry change during the phase transition process (Fig. 4). At temperatures above the phase transition, the SHG intensities were almost vanishing, consistent with a centrosymmetric structure. Upon cooling across the transition temperatures, the SHG intensities increase abruptly, indicating a change from centrosymmetric to noncentrosymmetric space groups. The SHG response displayed clear thermal hysteresis, with transition temperatures in excellent agreement with those obtained from magnetometry. Therefore, the symmetry-breaking nature of this phase transition process was further confirmed.

Metastable states often exhibit distinct physical properties compared to those of the ground state and can be achieved through light irradiation and rapid cooling. Given that the low-temperature phase is polar, we propose that the metastable state could facilitate access to the paraelectric phase at low temperatures.

When the sample were directly placed at a temperature below 60 K from the HT phase, the $\chi_m T$ would be maintained at *ca.* 4.05 cm³ mol⁻¹ K, indicating the presence of a quenching effect (Fig. 2). This value was consistent with that of the HT phase, which suggests that the metastable state induced by quenching resembles the high-temperature phase. Further heating caused an abrupt decrease in the $\chi_m T$ to approximately 0.78 cm³ mol⁻¹ K above 120 K, through a relaxation process from a quenching (MS*) state back to the LT state at $T_{1/2}$ = 107 K. To further explore the structural rearrangements, we collected single-crystal structures of freshly filtered **1** at 30 K under quenching (MS*) conditions. Notably, the quenching effect stabilized the *P*2/*n* space group, indicating that the MS* structure closely resembles the HT phase (Table S3). The asymmetric unit comprised two half-trinuclear motifs, two water molecules, half a disordered EtOH molecule, and an ordered EtOH molecule. The Co–N bond lengths of both Co ions ranged from 2.066(2) to 2.149(2) Å, suggesting the typical coordination of $Co^{II}_{HS}$. The Fe–N and Fe–C bond lengths of both Fe ions ranged from 1.945(2) to 1.980(4) Å and 1.903(3) to 1.923(3) Å, attributable to two $Fe^{III}_{LS}$ ions (Table S4). The disordered EtOH molecule formed hydrogen bonds with the cyanide groups in the two trinuclear units that were adjacent to it (Fig. 2). The main structural difference between the HT phase and the MS* originated from the different orientations of the ordered EtOH molecules.

Rapid cooling of the sample preserves the paraelectric phase at low temperatures, rather than transitioning to the ferroelectric phase, indicating that ferroelectricity can be controlled through the quenching effect (Figs. 3, S4). During relaxation to the low-temperature phase, a ferroelectric phase transition occurs from the *P*2/*n* (MS* phase) to the *Pn* (LT phase), implying that the paraelectric phase can be the thermodynamically metastable state.

In addition, photoinduced ETCST behavior of **1** was also detected. The MS phase was obtained by irradiating a fine, freshly filtered microcrystalline sample with a red light (785 nm, 10 mW) at 5 K (Fig. 1). Upon irradiation, the $\chi_m T$ rapidly increased with a saturation of *ca.* 4.05 cm³ mol⁻¹ K around 15 K after 30 min. This suggested that the MS formed with a practically quantitative light-excitation effect. A sudden decrease from *ca.* 4.05 to *ca.* 0.78 cm³ mol⁻¹ K was observed at $T_{1/2}$ = 84 K, suggesting a relaxation process from the MS back to the LT state.

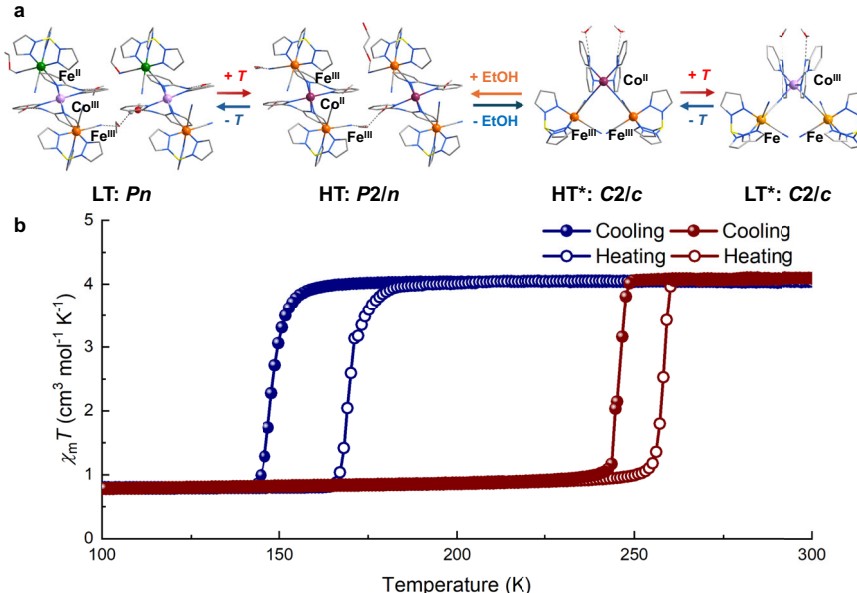

**Fig. 5 | Molecular structure, space group, and spin and valence states of the metal centers between 1 and 1′ during dynamic vapor sorption. a** Schematic representation of the on/off switching of ferroelectricity via desorption/adsorption of the guest molecule. The HT phase of 1 was in the nonpolar space group $P2/n$, the LT phase of 1 was in the polar space group $Pn$, while the LT* and HT* phases of 1′ were in the nonpolar space group $C2/c$. In the LT* phase, the two irons were in the average state of $Fe^{II}_{LS}$ and $Fe^{III}_{LS}$. The transition between the LT* and HT* phases was a temperature-dependent reversible process. Orange, $Fe^{III}$; green, $Fe^{II}$; purple, $Co^{II}$; pink, $Co^{III}$; yellow, B; gray, C; red, O; blue, N; light gray, H. **b** Temperature-dependent $\chi_m T$ product of 1 (blue) and 1′ (red). Open circle, heating process; filled circle, cooling process.

Light-induced ferroelectric phase transition has also been confirmed by single-crystal structural analysis (Fig. S4). After irradiation with 785 nm laser (10 mW) for 30 min, the compound underwent a transition from the ferroelectric phase ($Pn$) to the paraelectric phase ($P2/n$). The data showed that the structure in MS was similar to that in the HT phase and MS* state (Table S3), with the primary distinction among the three phases being the subtle differences in the orientations of the crystallization solvents, namely water and ethanol. The asymmetric unit comprised two half-trinuclear motifs, two water molecules, half a disordered EtOH molecule, and an ordered EtOH molecule. The Co–N bond lengths of both Co ions ranged from 2.074(4) to 2.159(3) Å, suggesting the typical coordination of $Co^{II}_{HS}$. The Fe–N and Fe–C bond lengths of both Fe ions ranged from 1.953(4) to 1.980(4) Å and 1.902(4) to 1.932(4) Å, attributable to two $Fe^{III}_{LS}$ ions (Table S4).

Through the quenching effect and light radiation, we successfully stabilized the paraelectric phase at low temperatures in this ferroelectric material by utilizing the metastable state. This achievement also demonstrates effective control over the ferroelectric phase transition through multistate switching.

The occurrence of a thermal-induced ferroelectric type phase transition indicates that the pristine sample is in the ferroelectricity "on" state. As shown in Fig. 3, in this state, there are two kinds of $[Fe(Tp)(CN)_3]_2[Co(dpa)_2]$–EtOH structural units labeled with red (A) and blue (B) circles in the HT and LT phases, respectively. In the HT phase, the ordered EtOH molecule formed a hydrogen bond with the neighboring cyanide, and the disordered EtOH molecule formed a hydrogen bond with two neighboring cyanides. After the transition from the LT to HT phase, the dipole moment of the trinuclear molecular unit changed via ETCST, where the orientation of EtOH in the A-unit changed from the ordered state in the LT phase to a disordered state in the HT phase. The EtOH in the A-unit was rearranged to form a hydrogen bond with the noncoordinating cyanide group of $Fe^{II}$. For the B-unit, the EtOH molecule is always in an ordered state with only slight orientation changes during the thermal-induced ETCST process. The structural analysis showed that the compound underwent a directional

ETCST during the structural transition in response to the temperature, where the EtOH molecules play an important role.

Given that ethanol molecules play a key role in the ferroelectric phase transition, we speculate that the compound will lose its ferroelectricity after the removal of ethanol. The process of ethanol removal was characterized by powder XRD (Fig. S1b) and thermogravimetric analysis (TGA, Fig. S5a). The TGA of 1 revealed a weight loss of 8.45% from 65 °C to 200 °C, corresponding to the loss of two water and one and a half EtOH molecules (8.75%), thus confirming the chemical composition. After the weight loss, a plateau was observed up to the point where decomposition occurred at *ca.* 200 °C, suggesting the formation of $[Fe^{III}(Tp)(CN)_3]_2[Co^{II}(dpa)_2]$. When the pristine sample was heated to 150 °C and then cooled, a gradual increase in weight was observed during the natural cooling process in the air, eventually stabilizing at a total weight loss of approximately 6% (Fig. S5b), which corresponds to a new dihydrate phase with the release of 1.5 EtOH molecules, 1′. To confirm the stability of this 1′ phase, we performed TGA on 1′ single crystals that had been stored at room temperature for one month (Fig. S5c). Upon heating, a weight loss of 3.18% was observed, consistent with the release of two water molecules. Notably, this weight loss was fully recovered during the subsequent natural cooling process, indicating that 1′ does not absorb more water molecules from ambient air under these conditions and remains structurally stable.

Interestingly, a solvent-dependent change in $\chi_m T$ was observed during the direct-current magnetometry (Fig. 5b). After removal of ethanol, the temperature-dependent magnetic susceptibility showed that 1′ underwent an ETCST process (the 1′ was obtained by exposing the fresh 1 sample to the air at room temperature for one month, which was confirmed by the powder XRD patterns). Below 230 K, the magnetic susceptibility and temperature ($\chi_m T$) value was maintained at *ca.* 0.85 cm³ mol⁻¹ K, which correlated with a $Fe^{III}_{LS}$ ion with $S = 1/2$, suggesting that the diamagnetic pair $[Fe^{II}_{LS}–Co^{III}_{LS}]$ was formed in the new low-temperature phase (LT*). Further heating led to an abrupt increase in the $\chi_m T$ to *ca.* 4.05 cm³ mol⁻¹ K above 262 K, which was consistent with the value obtained with an uncoupled $Co^{II}_{HS}$ ion ($S = 3/2$) and two

$Fe^{III}_{LS}$ ions ($S = 1/2$) in the new high-temperature phase (HT*). In the cooling process, the same transition of magnetic susceptibility occurred at a different temperature than in the pristine sample. By analyzing the $\chi_m T$ with respect to temperature, phase transition temperatures of $T_{1/2}(\downarrow) = 245$ K and $T_{1/2}(\uparrow) = 258$ K with a hysteresis of 13 K were obtained. Thus, another reversible bistability from the thermally induced ETCST between the LT* phase [$Fe^{II}_{LS}$–$Co^{III}_{LS}$–$Fe^{III}_{LS}$] and HT* phase [$Fe^{III}_{LS}$–$Co^{II}_{HS}$–$Fe^{III}_{LS}$] states was confirmed. This process was supported by variable-temperature IR spectra recorded from 250 to 300 K (Fig. S6).

After removal of ethanol, single-crystal structures of 1′ were collected at 120 K (LT* phase) and 300 K (HT* phase) to characterize the structural rearrangements during both thermally induced ETCST processes (Fig. 5a, Table S5). Notably, during the transformation from 1 into 1′, the space group changed from nonpolar $P2/n$ to nonpolar $C2/c$ in the HT phase. The asymmetric unit comprised a half [Fe(Tp)(CN)$_3$]$_2$[Co(dpa)$_2$] trinuclear motif and two water molecules. The Co–N bond lengths of both Co ions ranged from 2.084(3) to 2.152(3) Å, suggesting the typical coordination of $Co^{II}_{HS}$. The Fe–N and Fe–C bond lengths of both Fe ions ranged from 1.986(3) to 1.992(6) Å and 1.909(3) to 1.935(4) Å, respectively, which were common values for $Fe^{III}_{LS}$ ions. The unit cell volume shrank because the distance of two trinuclear motifs significantly reduced after the EtOH desorption.

With a decrease in temperature, the crystals remain in the nonpolar $C2/c$ space group at the LT* phase. The asymmetric unit comprised a half [Fe(Tp)(CN)$_3$]$_2$[Co(dpa)$_2$] trinuclear motif and two water molecules, similar to the HT* phase. The Co–N bond lengths of both Co ions ranged from 1.887(2) to 1.945(2) Å, suggesting the typical coordination of $Co^{III}_{LS}$. The Fe–N and Fe–C bond lengths of both Fe ions ranged from 1.992(2) to 1.999(2) Å and 1.875(3) to 1.925(3) Å, respectively (Table S6). These structural characteristics suggested the existence of [$Fe^{III}_{LS}$–$Co^{III}_{LS}$–$Fe^{II}_{LS}$]. However, $Fe^{III}$ and $Fe^{II}$ could not be distinguished because of random occupation at different lattice sites[48,51,52]. This non-directional ETCST behavior prevents polarization switching at the crystal level, resulting in the ferroelectricity "off" state upon EtOH desorption (Fig. S7).

This ethanol removal process was observed to be reversible. After exposure to the vapor atmosphere of ethanol, the sample was recovered and exhibited the same thermal stability shown in the TGA curves. Consequently, this ethanol removal and absorption process can occur in the same crystal, and the reversibility of this process suggests that the sample underwent dynamic vapor sorption. Upon the absorption process, 1′ transforms to 1, thereby restoring the ferroelectric properties. This transition enables on/off switching of ferroelectricity through the reversible absorption and desorption of guest molecules.

A ferroelectric phase transition was observed during the thermally induced ETCST[30]. The structural analysis showed that during the thermally induced ETCST process, the orientation of EtOH molecules underwent directional changes with the emergence of distinguishable $Fe^{II}$, which implied the existence of a directional ETCST process at the single-crystal level. The molecular packing revealed a directional electron transfer between Fe and Co ions. This electron transfer process was accompanied by a transformation between the disordered and ordered states of the EtOH molecule (Figs. 2, S8). This is tentatively ascribed to the change in the dipole of the trinuclear molecule after the directional electron transfer, leading to a directional rotation of the EtOH molecule.

To provide more direct evidence of polarization change, a single-crystal sample was sandwiched with silver pastes on the (001) and (00–1) surfaces for pyroelectric measurement. For the ferroelectric phase transition process from HT to LT, during the cooling process, the peak of the pyroelectric coefficient ($p$) was observed at $T(\downarrow) = 148$ K, which is consistent with the magnetic properties and structural changes during the phase transition from HT to LT, indicating a polarization change (Fig. S9). The polarization change ($\Delta P$)

obtained by integrating $p$ is about 140 nC/cm during the temperature decrease from 170 to 130 K, which is almost same with the polarization change during the relaxation process from MS* to LT, indicating that the ferroelectric phase transition process is reliable. Noting that crystals tend to crack during the ETCST process, it is difficult to observe reversible polarization changes of a single crystal sample during the heating and cooling processes in the experiments. To further demonstrate the ferroelectric transition, the temperature-dependent polarization change was obtained by integrating the pyroelectric current under a reverse electric field based on a powder pellet of 1 (Fig. S10). The results of reversed spontaneous polarization under a direct-current electric field confirmed the ferroelectricity of 1. Thus, the thermally induced polarization switching is well characterized, indicating a thermo-induced ferroelectric phase transition.

Noting the quenching effect of 1, we placed the crystal sample directly at 30 K from room temperature (300 K) and recorded the current signal during the heating process to verify the occurrence of the ferroelectric phase transition. During the thermal relaxation process, the peak of pyroelectric coefficient ($p$) was observed at $T(\uparrow) = 118$ K, which is consistent with the magnetic properties and structure change during the phase transition from MS* to LT, indicating a polarization change (Fig. 6). The polarization change($\Delta P$) obtained by integrating $p$ is about 120 nC/cm$^2$ during the temperature increased from 50 to 150 K, which is consistent with the thermally induced ferroelectric phase transition process. Additionally, the polarization change during the photoinduced ferroelectric phase transition was detected at the single-crystal level. Upon irradiation with a 785 nm laser at 5 K, a sharp current peak was immediately observed, indicating a transition from the polar phase to a nonpolar phase. The light exposure was maintained for 500 s, during which the current gradually decreased to a negligible level (Fig. S11). Subsequently, the temperature was increased. As the photoinduced nonpolar MS state relaxed back to the polar LT state, a reverse peak in the pyroelectric current was observed at approximately 110 K, confirming the recovery of the polar phase.

To theoretically investigate the origin of polarization change during the structural transition, point charge and density functional theory (DFT) calculations were conducted (Fig. 7a, b). The point charge calculations suggested that the polarization change caused by the pure electron transfer behavior along the $c$-axis was 2.45 μC cm$^{-2}$ (supporting information). The DFT calculations were performed on the asymmetric unit that comprised two trinuclear units and three EtOH molecules, which were extracted from the LT X-ray structure with optimized positions of hydrogen atoms. Given the symmetry constraints, spontaneous polarization was expected to occur in the $ac$ plane. To distinguish the contributions of electron transfer and EtOH reorientation to the polarization change, polarization was calculated for the models containing only the two trinuclear units or the three EtOH molecules, each held at fixed positions. The results showed that electron transfer contributed 3.22 μC cm$^{-2}$, whereas the reorientation of EtOH molecules contributed 0.43 μC cm$^{-2}$, with an angle of 160.4°. Thus, we concluded that directional electron transfer played a major role in the observed polarization change.

To summarize the above results, we observed that, in this crystal, polarization can be controlled through desorption/absorption of guest molecules and metastable state via quenching effect and light irradiation. During rapid cooling, MS* enables retention of the high-temperature paraelectric phase at low temperatures due to the quenching effect. In contrast, during slow cooling, the spontaneous polarization from the ferroelectric phase transition can be recorded via the current signal, thereby achieving polarization control via the cooling rate. After removal of ethanol, a single-crystal to single-crystal transformation occurred; however, no ferroelectric phase transition

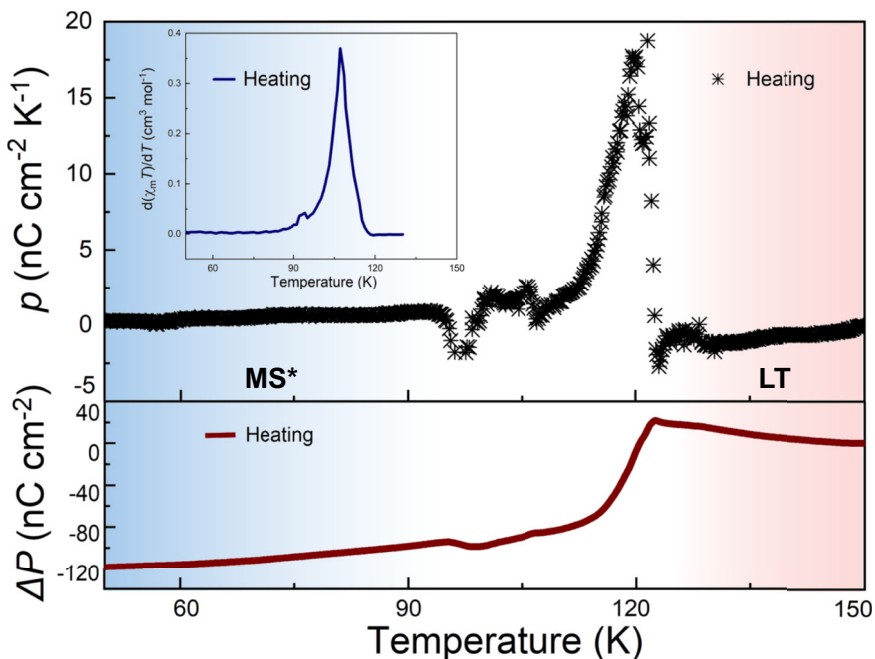

**Fig. 6 | Temperature dependence of the pyroelectric coefficient ($p$) and polarization change ($\Delta P$) plot.** Pyroelectric properties of 1 were observed by directly placing the crystal sample at 30 K and then recording the changes from quenching state (MS*) to low-temperature phase (LT) during the heating process. Inset: first-order derivatives of the $\chi_m T$ product.

was observed in the new phase, **1'**. Thus, polarization control through desorption/adsorption of guest molecules was also achieved, which demonstrates that the orientation-directed change of the lattice solvent facilitates directional ETCST at the crystal level.

The results of this study demonstrate that six stable states—HT, LT, HT*, LT*, MS, and MS*—can be achieved in **1** complex via guest molecule-dependent, thermally induced, and photo-induced ETCST processes. These states and their stability have been demonstrated via single-crystal structural analyses, magnetometry, and IR spectroscopy. During the ETCST process, the directional electron transfer between Fe and Co, accompanied by the rearrangement of the EtOH molecules within the crystal lattice, induces polarization switching, representing an example of polarization control via ETCST. The observed ferro-electric behavior, driven by intramolecular electron transfer, high-lights the promise of cyanide-bridged compounds as candidates for next-generation memory devices. Moreover, the ability to control the polarization through both cooling rate and desorption/absorption of the guest molecules introduces new strategies for manipulating the spontaneous polarization of ferroelectrics. The reversible on/off switching of ferroelectricity via the dynamic desorption/adsorption of guest molecules offers a promising new approach to mitigate ferro-electric fatigue. This study builds upon our previous findings, further validating the feasibility of directional ETCST at the single-crystal level, providing valuable insights to guide the design and synthesis of additional electron transfer compounds with polarization switching capabilities[53–58]. Our light-induced polar-to-non-polar conversion, combined with guest-controlled on/off ferroelectricity, complements prior light-assisted polar-state control and establishes a distinct axis for photo-programmable ferroics. We anticipate that coupling ETCST to lattice symmetry will enable new device concepts for non-volatile, optically addressable states[59,60].

## Methods

### Materials and synthesis methods
All chemicals used were of analytical grade, and solvents were kept anhydrous using molecular sieves. The dpa was commercially available (dpa = 2,2′-dipyridylamine).

### Synthesis of [Fe(II)(Tp)$_2$]
FeCl$_2$·6H$_2$O (59 g, 0.25 mol) was added to a solution of KTp (126 g, 0.5 mol) under vigorous stirring. A purple precipitation of Fe(II)Tp$_2$ was formed, which was collected by filtration and dried in the air. Yield: 99% (119 g).

### Synthesis of Bu$_4$N[Fe(Tp)(CN)$_3$]
[Fe(II)(Tp)$_2$] (48 g, 0.1 mol) and NaCN (25 g, 0.5 mol) were mixed in 2-propanol (300 mL), and the resulting suspension was heated at 80 °C for five hours under continuous stirring. The initial purple solid turned yellowish brown and was poured off while leaving unreacted NaCN at the bottom. The solid was washed with hot water, air dried, dissolved in 500 mL of water, and filtered again to remove undissolved materials. A solution of Bu$_4$NBr (32.2 g, 0.1 mol) was added to the filtrate under stirring, with formation of a brownish solid. Dropwise addition of 30% H$_2$O$_2$ to this suspension caused precipitation of the crude product as an orange-red crystalline solid. Yield: 85% (48 g).

### Synthesis of [Fe(Tp)(CN)$_3$]$_2$(Co(dpa)$_2$)$_2$·2H$_2$O·1.5EtOH (1)
An aqueous solution of Co(II)(ClO$_4$)$_2$·6H$_2$O (36.5 mg, 0.1 mmol, 2.5 mL) was placed at the bottom of a long test tube followed by a buffer layer of 0.6 ml EtOH and 0.4 ml water, and a solution of Bu$_4$N[Fe(Tp)(CN)$_3$] (112 mg, 0.2 mmol) and dpa (34 mg, 0.2 mmol) in 2.5 mL EtOH was layered over it. This was left undisturbed at 40 °C for one month, and light red prism crystals were separated by filtration and washed with water (3 mL × 3). The yield was about 36% based on Co(II)(ClO$_4$)$_2$·6H$_2$O. (Elemental analysis calcd (%) for C$_{47}$H$_{51}$N$_{24}$O$_{3.5}$B$_2$Fe$_2$Co: C 47.03, H 4.28, N 28.01; Found: C 47.11, H 4.31, N 28.11)

### Synthesis of [Fe(Tp)(CN)$_3$]$_2$(Co(dpa)$_2$)$_2$·2H$_2$O (1′)
The ethanol-removed sample **1'** was obtained by exposing the fresh **1** sample to the air at room temperature for one month.

### X-ray crystallography
All single crystals were coated with an oil-based cryoprotectant and mounted on nylon loops. Single-crystal X-ray structures were obtained for all phases. The Crystallographic Information Files of the crystal

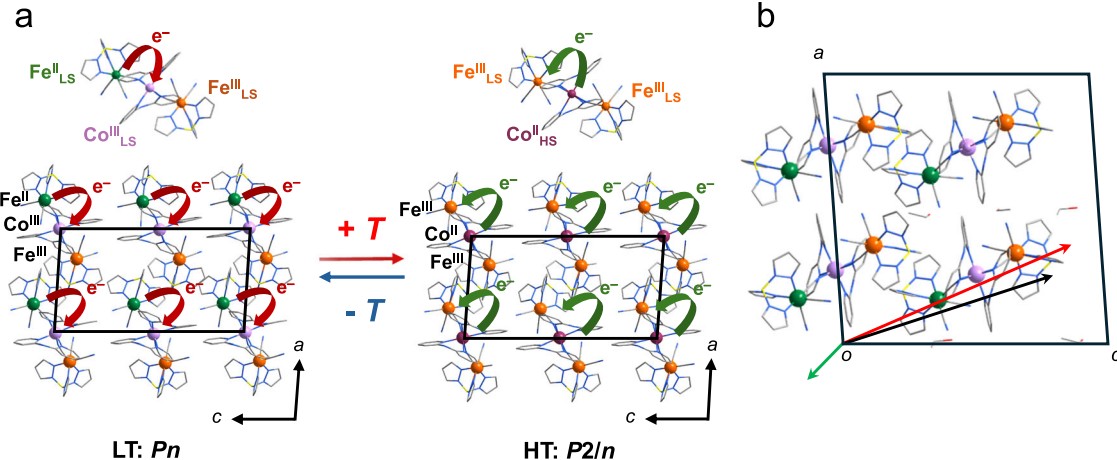

**Fig. 7 | Directional ETCST induced ferroelectric phase transition and polarization switching mechanism. a** Schematic representation of **1** in the ferroelectricity "on" state via directional ETCST. The LT phase was in the polar space group *Pn*, whereas the HT phase was in the nonpolar space group *P*2/*n*. During the phase transition, the compound underwent the directional ETCST. Orange, Fe$^{III}$; green, Fe$^{II}$; purple, Co$^{II}$; pink, Co$^{III}$; yellow, B; gray, C; red, O; blue, N. Red arrow, direction of electron transfer; yellow arrow, polarization direction within the crystal lattice. **b.** Calculated polarization change from the directional charge transfer and EtOH reorientation. Green arrow, contribution from EtOH reorientation (0.43 μC/cm²); Red arrow, contribution from directional electron transfer (3.22 μC/cm²); Black arrow, net polarization change (2.45 μC/cm²).

structures can be obtained free of charge from the Cambridge Crystallographic Data Center for **1** at 100 K (LT) and 300 K (HT), for **1'** at 120 K (LT*) and 300 K (HT*) (CCDC Rigaku FR-E+ diffractometer equipped with a HyPix-6000 area detector, using a multilayer mirror monochromated Mo-$K\alpha$ radiation ($\lambda = 0.71073$ Å), for **1** after irradiation (MS) and quenching (MS*) at 30 K (CCDC Rigaku varimax diffractometer equipped with a HyPix-6000 area detector, using a multilayer mirror monochromated Mo-$K\alpha$ radiation ($\lambda = 0.71073$ Å)). The structures were solved by a direct method and refined via full-matrix least-squares on $F^2$ using the SHELX program[61,62] implemented in the OLEX2 program[63] with anisotropic thermal parameters for all non-hydrogen atoms. The hydrogen atoms were geometrically added and refined by the riding model. Powder X-ray diffraction patterns were recorded on a Rigaku-TTR diffractometer to examine purity and the difference between **1** and **1'**.

### Magnetism measurements

Magnetic susceptibility measurements were conducted on a Quantum Design MPMS-XL superconducting quantum interference device magnetometer under a 2000 Oe field, with a sweeping rate of 2 K·min⁻¹ in a 5–300 K temperature range. The samples were prepared by encapsulating polycrystalline samples (ca. ~15 mg) into a gelatin capsule. Photomagnetic measurements were performed on powdered samples attached to transparent tape. A 785 nm laser (10 mW) was used as the excitation source. The sample was irradiated for 1 h with the cooling valve open to maintain a low-temperature environment. Relaxation measurements were performed from 5 to 130 K.

### IR spectroscopy

Temperature-dependent IR spectra were recorded on a JASCO FT/IR-600 Plus spectrometer equipped with a temperature control system. The powder samples were held between grained and flat CaF₂ plates. The experiment of **1** was performed with the temperature ranging from 140 to 200 K in the heating process. The experiment of **1'** was performed with the temperature ranging from 100 to 300 K in the heating process.

### Optical second-harmonic generation

Optical second-harmonic generation (SHG) was measured using a femtosecond Er³⁺-doped fiber laser (1550 nm, 100 fs, 20 MHz; Calmar FPL-03C). Thin flakes from a crushed single crystal were mounted in a heat-exchange-gas optical cryostat for continuous temperature control. A chopped excitation beam was normally incident, and the transmitted SHG signal was isolated by multiple optical filters, detected with a cooled photomultiplier, and analyzed by lock-in detection.

### Theoretical calculations

DFT calculations were conducted using the ORCA 5.0.1 quantum chemistry package[64,65]. An asymmetric pair consisting of [Fe₂Co] and ethanol molecules was extracted from the single-crystal structure recorded at 100 K, with the Cartesian *x*- and *y*-axes aligned to the crystallographic *a* and *b*-axes, respectively. The *z*-axis was defined as the cross product of the *x*-vector and *y*-vector, and this orientation remained fixed throughout the process. The coordinates of all non-hydrogen atoms were kept fixed, while only the hydrogen atoms' coordinates were optimized geometrically. To prevent self-interaction errors that could lead to a mixing of the two Fe sites' valence, the range-separation functional cam-B3LYP[66–70] was employed for all calculations, using the Karlsruhe triple-zeta basis def2-TZVP[71–73]. Atom-pairwise dispersion correction was applied along with Becke–Johnson damping[74,75]. The molecular dipole moments were then calculated based on structural fragments extracted from the optimized model. The macroscopic polarization was determined as the volume average of the sum of the molecular dipole moments in a unit cell.

## Data availability

All data to support the conclusions are available in the main text or the supplementary materials. CCDC 2352581 (LT*), 2352582 (HT*), 2352583 (MS*), 2352585 (MS), 2352586 (HT), and 2377474 (LT) containing the supplementary crystallographic data can be obtained free of charge from the Cambridge Crystallographic Data Center via www.ccdc.cam.ac.uk/data_request/cif. The magnetic, electrical, spectroscopic data and optimized molecular geometry used in this study are available in the QIR database under accession code https://hdl.handle.net/2324/7399677. All data are available from the corresponding author upon request. Source Data are provided with this manuscript. Source data are provided with this paper.

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

## Acknowledgments

This work was supported by the MEXT Project of "Integrated Research Consortium on Chemical Sciences". O. Sato thanks the JSPS KAKENHI (24H00466). S.-Q. Wu is grateful for support from JSPS KAKENHI (24K17698) and Murata Science and Education Foundation (M24AN117). K.Y. acknowledges the JSPS KAKENHI (25K08602). Y.-B. Huang thanks the China Scholarship Council for support. Dr. Hiroyasu Sato is acknowledged for his assistance in the single-crystal structural analysis. Prof. Ken Kojio is acknowledged for his measurements of differential scanning calorimetry.

## Author contributions

Y.H., S.S., and O.S. designed the study, conducted experiments, and wrote most of the paper. S.W. performed DFT calculations, assisted in the electric measurements and revised the manuscript. W.X., W.Z., T.J., H.C., and Z.L. assisted in structural measurements and results analysis. K. G. assisted in pyroelectric measurements. S.I. and K.Y. performed SHG measurements. All authors discussed the results and commented on the manuscript.

## Competing interests

The authors declare no competing interests.
