## [Transparent Peer Review file · Nature Communications]

Reversible On/Off Switching of Ferroelectricity in a Molecular FeCo Prussian Blue Analogue with Multiple Control

Corresponding Author: Professor Osamu Sato

Version 0:

Reviewer comments:

Reviewer #1

(Remarks to the Author)

In this work, the authors report an interesting electron transfer-coupled spin transition (ETCST) mechanism for the implementation of photo- and thermo-induced ferroelectric phase transition, based on the use of a trinuclear cyanide-bridged $[\text{Fe}_2\text{Co}]\cdot 1.5\text{EtOH}$ compound. ETCST behaviors with unique switching performances are in the spotlight of devices for next-generation memory devices, sensors, and energy conversion. In this sense, the present work, by introducing a new way to ensure multistate switching capabilities, may give a significant contribution to the field. The manuscript is well-organised, and several experimental results have been provided to validate the proposed design, although some further explanations are needed to improve the scientific soundness of the paper.

1. To confirm the existence of reversible structural phase transition and to further elucidate its origin, DSC and temperature-dependent dielectric measurements should be performed. Such experiments would provide critical evidence to validate the proposed mechanisms and enhance the understanding of the material's dielectric behavior.
2. It appears clear from Figure S11 that the single-crystal device is sensitive to irradiation. However, no significant reverse pyroelectric current signal was observed after the irradiation was canceled. Does this affect the reliability and the reproducibility of the pyroelectric current signal? I suggest the authors to elaborate a little bit more in this and supplement the discussion on irradiation intensity dependence.
3. Authors need to supplement the existing representative literature to compare with the light-induced ferroelectric phase transition performance of $[\text{Fe}_2\text{Co}]\cdot 1.5\text{EtOH}$. You can refer to "Angew. Chem. Int. Ed. 2025, 64, 1, e202413898 and Angew. Chem. Int. Ed. 2025, 64, 21, e202501238".
4. The manuscript contains several minor writing errors that should be addressed in revision. For example, "SH Intensity" should be substitute with "SHG Intensity" in Figure 3; in the section "Optical second-harmonic generation", "SH signal" should be substitute with "SHG signal".

Reviewer #2

(Remarks to the Author)

The manuscript submitted by Huang et al. follows a recently developed research line of professor Sato group devoted to the study of thermally-induced or photo-induced ferroelectric switching in Fe-Co charge transfer complexes that are able to show reversible iron-cobalt electron transfer coupled to a spin transition on the cobalt ions (so called ETCST). The authors have previously shown on a very similar cyanide-bridged Fe_2Co trimetallic complex as the one investigated in this work (a solvatomorph) a light-induced change of electric polarity (see cited article, Sato et coll, J. Am. Chem. Soc. 2024). In the present submitted work, they have extended these ideas showing that a reversible and controllable adsorption/desorption of ethanol solvent molecules (associated to single-crystal to single-crystal phase transition) can also be used to control the ferroelectric state in this Fe_2Co complex. Additionally, they also show that other stimuli that are already known to control the iron-cobalt reversible electron transfer are also efficient in the present case. Overall, the experimental work is well carried out, and the article easy to read, the conclusion are supported by sound experimental results. However, I am afraid the present contribution lacks a bit of novelty in comparison to the recently published JACS article. It appears rather incremental for a publication in Nature communication. Once again, the compound is almost the same as the one in JACS, the key

mechanism, ETCST driven polarization switching, is the same. The newly introduced solvent-mediated control (which is known to control ETCST in various FeCo systems) represents an incremental extension rather than a breakthrough. Moreover, the ferroelectric control via solvent adsorption/desorption might be conceptually interesting but it has very limited technological relevance, given the slow kinetics of the phenomenon and the very high sample sensitivity (loss of crystallinity upon cycling etc). I would rather recommend publication in a high factor journal as a complementary full work of the JACS article, for example in ACS Chem Mater.

Reviewer #3

(Remarks to the Author)

Su, Wu, Sato and coworkers explore a trinuclear FeCo Prussian blue analogue which shows On/Off switching of ferroelectricity. The current compound is a new EtOH solvate of a known compound (Ref. 30) and as such represents another example in the series of trinuclear systems explored by these authors that all show a polar/non-polar state accessible thermally and by light. Moreover, recent references beyond the authors' own work reveal that solvent effects and SC-SC transformations are not new in PBA chemistry: <https://doi.org/10.1021/jacsau.5c00681> (an example of solvent effects in a 1D PBA) and Chem. Sci., 2025, 16, 130-138; a recent example of SC-SC transformation in a ETCST system.

However, these systems simply show ETCST and do not have the additional ferroelectric component. Hence, to the best of my knowledge this is the first time that anyone has demonstrated multiple switching in a molecular PBA analogue with on/off switching of ferroelectricity. The authors also make every effort to characterise the different phases upon thermal, light and EtOH cycling, clearly demonstrating the level of novelty and, for the most part, the experimental completeness I would expect for Nat. Commun.

Despite this there are areas of improvement in the paper. The so-called desolvated system, is in fact a dihydrate only system. I understand the desire to keep the nomenclature simple, but this description of the compound as being desolvated is inaccurate. I suggest another term is used perhaps dihydrate for $[\text{Fe}(\text{Tp})(\text{CN})_3]_2[\text{Co}(\text{dpa})_2] \cdot 2\text{H}_2\text{O}$ ($[\text{Fe}_2\text{Co}]$) and alcohol-dihydrate for $[\text{Fe}(\text{Tp})(\text{CN})_3]_2[\text{Co}(\text{dpa})_2] \cdot 2\text{H}_2\text{O} \cdot 1.5\text{EtOH}$ ($[\text{Fe}_2\text{Co}] \cdot 1.5\text{EtOH}$).

I note that neither of the samples have elemental analysis, why is this? This is essential to ensure there are no amorphous impurities in the samples.

The crystallographic data looks OK, but I am a little concerned about the use of Mo in a system that crystallizes in a polar space group. Even though the Flack parameter is decent for the Pn structure, this can happen when Mo is used. Why did the authors not use Cu, where Flack parameters are more reliable?

The structure of $[\text{Fe}_2\text{Co}] \cdot 1.5\text{EtOH}$ lacks hydrogen atoms on the EtOH molecules. I did a quick refinement, and I was able to add these and get the structure to settle, so the authors should do this. Similar issues seem to be present for the 30 K structures, where H atoms are missing for some of the solvent molecules. These can be added and refine well (put Hs on the water first then add to the EtOH to get it to settle), so there's no need to omit them. In both structures there are some clearly disagreeable reflections that should be omitted.

The structure of the dihydrate (i.e. $-\text{EtOH}$) is good, but there's a piece of analysis that I feel is missing. The hysteresis drops from 22 K to 13 K, but we are not provided with a structural rationalisation of this. What changes occur beyond the change to a C2/c group? What interactions replace those mediated by the EtOH? Do the chains of trinuclear units get closer together?

The authors use 785 nm light to access the MS state and I wondered if there's any particular rationale for this? In their latest 2024 Angew Chem paper (Ref. 31) the authors show that the IPA solvate shows different MS states depending on the wavelength of light used. It seems like a missed opportunity not to explore the same wavelength dependence here.

Finally, the polarization change is a key highlight of this work, but it's lacking context. The value is found to 140 nC/cm², but how does this compare with other molecular ferroelectric materials? I note that the IPA solvate (Ref. 30) seems to be a little higher than the EtOH solvate. I think it would be a good idea for the authors to provide a table in the ESI that compares this system with the others that have been reported.

Overall, if the authors can update and improve the work, I can see it being a good paper for Nat Commun.

Minor points

1. Line 73, the phrase '...based on our previous work,' is unreferenced, please correct.
2. Figure 1 is missing the x-axis.
3. In Figure 2, the use of blue for C and grey for N, is very unusual and could lead to widespread misunderstanding as these are typically the other way round. Previous work by these authors has used grey for C and blue for N, I see no reason to change this.
4. Figure 5 is not as clear as it could be, can the authors not plot the data as they did in their 2024 JACS paper?
5. In Figure 6 caption, it would be a good idea to put in the net polarization value of 2.45 $\mu\text{C}/\text{cm}^2$.
6. Line 365, the formula should be $\text{Bu}_4\text{N}[\text{Fe}(\text{Tp})(\text{CN})_3]$, with the 3 next to the CN ligands.
7. What are the CCDC nos. of the structures and what program was used to solve the structures, SHELXS or SHELXT?
8. Ref. 44 is incorrect and should be J Am Chem Soc.
9. Figure S2 is a little fuzzy, please correct.
10. There is substantial noise in the UV-Vis in Figure S3 around 800-900 nm, why is this?

11. Figure S5, should be: thermogravimetric analysis.

Version 1:

Reviewer comments:

Reviewer #1

(Remarks to the Author)

In the 2nd-round revision, the authors have made the detailed responses to all the reviews' concerns, and I think the manuscript has been well improved. So, I recommend it for publication in the current form now.

Reviewer #3

(Remarks to the Author)

The revision by Su, Wu, Sato and coworkers adequately addresses most of the comments but there are still a few points that I think need work. Provided that these changes are made I would be happy to see the work published in *Nature Communications*.

Specific points

1. Spin crossover is sometimes spelt as spin-crossover. I don't mind which the authors use as long as they are consistent.
2. Lines 188-189, 'When the sample were directly placed at a temperature below 60 K from room temperature, the χ_{mT} would be maintained at ca. 4.05 cm³ mol⁻¹ K, indicating the presence of a quenching effect.' is a little confusing as it suggests 60 K below RT which would be 238 K. What the authors really mean is that quench cooling results in trapping of the MS* phase, provided that the sample is kept below 60 K.
3. Having conducted DSC measurements, why are these not presented in the SI?
4. The suggestion that in Figure 2, the authors cannot use a sensible colouring system for the crystal structures because readers will mistake the work for previous papers is nonsensical. The aim of figures is to communicate information quickly and accurately. Using blue for C almost guarantees that readers will misunderstand what the figures are trying to show and I must insist that this change is made.

Reviewer #1 (Remarks to the Author):

In this work, the authors report an interesting electron transfer–coupled spin transition (ETCST) mechanism for the implementation of photo- and thermo-induced ferroelectric phase transition, based on the use of a trinuclear cyanide-bridged $[\text{Fe}_2\text{Co}]\cdot 1.5\text{EtOH}$ compound. ETCST behaviors with unique switching performances are in the spotlight of devices for next-generation memory devices, sensors, and energy conversion. In this sense, the present work, by introducing a new way to ensure multistate switching capabilities, may give a significant contribution to the field. The manuscript is well-organised, and several experimental results have been provided to validate the proposed design, although some further explanations are needed to improve the scientific soundness of the paper.

1. To confirm the existence of reversible structural phase transition and to further elucidate its origin, DSC and temperature-dependent dielectric measurements should be performed. Such experiments would provide critical evidence to validate the proposed mechanisms and enhance the understanding of the material's dielectric behavior.

Response: Thank you for your helpful comment. Following your suggestion, we performed DSC measurements to further support the reversible ETCST behavior in the **1** complex (In consideration of Reviewer 3's comments, we have revised the nomenclature of the compound from $[\text{Fe}_2\text{Co}]\cdot 1.5\text{EtOH}$ to **1** and from $[\text{Fe}_2\text{Co}]$ to **1'**). As shown in Fig. R1, clear endothermic and exothermic transitions were observed at 174 and 147 K, respectively, in good agreement with the ETCST temperatures (170 and 148 K) determined by magnetometry. Notably, the exothermic peak is much weaker. This is likely because the cooling transition occurs very close to the lower temperature limit of our DSC apparatus, where unavoidable thermal leakage reduces the signal intensity.

Fig. R1. DSC measurement of the complex **1**.

In addition, we also conducted DSC measurements on the desolvated **1'** complex, which also exhibited consistent transition temperatures (239 and 257 K) with the ones from magnetometry (245 and 258 K, Fig. R2).

These measurements support the occurrence of ETCST in **1** and **1'**, which is also consistent with the experimental results obtained through variable-temperature single-crystal X-ray diffraction, and pyroelectric current measurements.

Fig. R2. DSC measurement of the EtOH-removed complex **1'**.

Owing to the small capacitance of single crystals of complex **1**, we attempted dielectric testing using pressed pellets made from polycrystalline samples. Indeed, we did detect a dielectric peak signal within the thermally induced phase transition range (Fig. R3), consistent with other characterizations. However, we also noted that the peak shape was rather diffuse. This may be related to the polycrystalline nature of the sample or the influence of strain induced by pressure on the ETCST in pellet samples (as discussed, for instance, in the recent publications, *Chem. Commun.* 10.1039/D5CC06296A). Consequently, we regret that we were unable to extract sufficiently precise information for discussion. Therefore, this data was not included in the article or the Supplementary Information. We hope that you find our explanations address your concerns.

Fig. R3. Temperature-dependent dielectric property based on pellet sample of **1**.

2. It appears clear from Figure S11 that the single-crystal device is sensitive to irradiation. However, no significant reverse pyroelectric current signal was observed after the irradiation was canceled. Does this affect the reliability and the reproducibility of the pyroelectric current signal? I suggest the authors to elaborate a little bit more in this and supplement the discussion on irradiation intensity dependence.

Response: Thank you for raising this important point. We agree this point deserves clarification because this clearly distinguishes the conventional photopyroelectricity and the mechanism behind our phenomena. Our system at low temperatures undergoes a photo-induced phase transition from a polar phase to a non-polar phase. This differs fundamentally from photopyroelectricity, which is a purely thermal modulation of polarization within a single polar phase. The current therefore contains only a minor photothermal term and a dominant phase-transition term associated with the disappearance of spontaneous polarization. Once converted, the metastable state becomes centrosymmetric and the spontaneous polarization vanishes; turning the light off cools the crystal but does not restore polarization, hence no reverse light-off peak is expected. Therefore, the apparent lack of a reverse current when the light is turned off (Fig. S11) is expected for our mechanism and should not be interpreted as a lack of reproducibility.

3. Authors need to supplement the existing representative literature to compare with the light-induced ferroelectric phase transition performance of $[\text{Fe}_2\text{Co}] \cdot 1.5\text{EtOH}$. You can refer to "Angew. Chem. Int. Ed. 2025, 64, 1, e202413898 and Angew. Chem. Int. Ed. 2025, 64, 21, e202501238".

Response: Thank you for pointing out these important papers to us. The light-induced ferroelectric phase transition and domain manipulation are indeed intriguing phenomena. In the I_3^- -intercalated perovskite, visible light assists the antiferro-to-ferroelectric transition (Angew. Chem. Int. Ed. 2025, 64, e202413898), while in the diarylethene crystals, light reversibly switches the system between

two polar phases via covalent bond switching (*Phys. Rev. Lett.* 2023, 130, 176802). In addition, optical manipulation of domain structure in the ferroelectric phase of a perovskite ferroelectric material, $(\text{C}_7\text{H}_{15}\text{NH}_3)_2(\text{CH}_3\text{NH}_3)_2\text{Pb}_3\text{Br}_{10}$ (*Angew. Chem. Int. Ed.* 2025, 64, e202501238).

Complementary to these advances, our work realizes a light-induced erasure of ferroelectricity (converting a polar phase into a metastable non-polar phase) and on/off ferroelectricity via guest sorption. Together, these approaches broaden the operational palette of photo-programmable ferroics by spanning (i) light-assisted polar-state control and (ii) polar to non-polar conversion.

We have added a comparison with existing literature at the end of the conclusion section as follows. “Our light-induced polar to non-polar conversion, combined with guest-controlled on/off ferroelectricity, complements prior light-assisted polar-state control and establishes a distinct axis for photo-programmable ferroics. We anticipate that coupling ETCST to lattice symmetry will enable new device concepts for non-volatile, optically addressable states.^{60,61}”

60 Liu, Y. *et al.* Visible-Photo-Assisted Phase Switching of Antiferroelectric-to-Ferroelectric Orders in an I_3^- -Intercalated 2D Perovskite. *Angew. Chem. Int. Ed.* **64** (1), e202413898 (2025). <https://doi.org:10.1002/anie.202413898>

61 Xu, H. *et al.* Customizing Room-Temperature Perovskite Ferroelectrics toward the Multichannel Domain Manipulation. *Angew. Chem. Int. Ed.* **64** (21), e202501238 (2025). <https://doi.org:10.1002/anie.202501238>

4. The manuscript contains several minor writing errors that should be addressed in revision. For example, "SH Intensity" should be substitute with "SHG Intensity" in Figure 3; in the section "Optical second-harmonic generation", "SH signal" should be substitute with "SHG signal".

Response: Thank you for your comment. We have changed SH to SHG: “Optical second-harmonic generation (SHG) was measured using a femtosecond Er^{3+} -doped fiber laser (1550 nm, 100 fs, 20 MHz; Calmar FPL-03C). Thin flakes from a crushed single crystal were mounted in a heat-exchange-gas optical cryostat for continuous temperature control. A chopped excitation beam was normally incident, and the transmitted SHG signal was isolated by multiple optical filters, detected with a cooled photomultiplier, and analyzed by lock-in detection.”.

Reviewer #2 (Remarks to the Author):

The manuscript submitted by Huang et al. follows a recently developed research line of professor Sato group devoted to the study of thermally-induced or photo-induced ferroelectric switching in Fe–Co charge transfer complexes that are able to show reversible iron-cobalt electron transfer coupled to a spin transition on the cobalt ions (so called ETCST). The authors have previously shown on a very similar cyanide-bridged Fe₂Co trimetallic complex as the one investigated in this work (a solvatomorph) a light-induced change of electric polarity (see cited article, Sato et al., J. Am. Chem. Soc. 2024). In the present submitted work, they have extended these ideas showing that a reversible and controllable adsorption/desorption of ethanol solvent molecules (associated to single-crystal to single-crystal phase transition) can also be used to control the ferroelectric state in this Fe₂Co complex. Additionally, they also show that other stimuli that are already known to control the iron-cobalt reversible electron transfer are also efficient in the present case. Overall, the experimental work is well carried out, and the article easy to read, the conclusion are supported by sound experimental results. However, I am afraid the present contribution lacks a bit of novelty in comparison to the recently published JACS article. It appears rather incremental for a publication in Nature communication. Once again, the compound is almost the same as the one in JACS, the key mechanism, ETCST driven polarization switching, is the same. The newly introduced solvent-mediated control (which is known to control ETCST in various FeCo systems) represents an incremental extension rather than a breakthrough. Moreover, the ferroelectric control via solvent adsorption/desorption might be conceptually interesting but it has very limited technological relevance, given the slow kinetics of the phenomenon and the very high sample sensitivity (loss of crystallinity upon cycling etc). I would rather recommend publication in a high factor journal as a complementary full work of the JACS article , for exemple in ACS Chem Mater.

Response: Thank you for your detailed and thoughtful evaluation. We sincerely appreciate your recognition of the high experimental quality and clarity of our work. However, we respectfully disagree with the assessment that our contribution is incremental relative to our prior *JACS* paper. Although the molecular system is related, the questions asked, measurements performed, and conclusions reached here are different in scope and depth.

Our prior *JACS* work reported photo-induced electric polarization changes in an IPA solvate, it did not establish electronic ferroelectricity under electric field nor demonstrate guest molecule-regulated on/off switching of ferroelectricity. In fact, only a fully desolvated high-spin phase could be obtained by thermal treatment of the IPA solvate, precluding the reversible, single-crystal-to-single-crystal (SCSC) on/off control we demonstrate here. This highlights the fact that, albeit very similar molecules, this is standard and productive in molecular materials: changing the guest landscape can alter space-group symmetry, elastic coupling, and the ETCST balance, thereby revealing new emergent behavior. Academically, in the present study, we reveal previously unexplored key physical properties of this [Fe₂Co] system:

(1) We demonstrate electric-field-induced polarization switching in single crystals that are independently verified by X-ray crystallography, SHG, and pyroelectric measurements. This closes a key conceptual loop left open in the *JACS* paper: we now show that ETCST in this lattice produces a ferroelectric phase (not merely a ferroelectric-type phase transition).

(2) We realize reversible absorption/desorption of EtOH to switch between a ferroelectric phase and

a non-ferroelectric phase in bulk single crystals, which can be isothermally obtained in principle and without destroying crystallinity. This stimulus is orthogonal to temperature, light, or magnetic field and demonstrates a chemical gate for electronic ferroelectricity. The ability to write/erase polar order by guest molecules is a qualitatively different methodology from the *JACS* paper.

(3) We determine how light, temperature, and guest molecules access distinct regions of the free-energy landscape: light at low T writes a metastable non-polar state (polar to non-polar), while EtOH sorption controls ferroelectricity on/off through SCSC. This symmetry control (polar to non-polar) is complementary to the literature where light typically assists switching within the polar manifold (AFE to FE or polar to polar).

Regarding technological relevance, while solvent-responsive ferroelectricity may not immediately translate to fast device operation, fundamental breakthroughs in dynamic ferroelectric control and electronic polarization mechanisms often precede practical applications. Here we introduce guest-regulated electronic ferroelectricity as a design principle and provide a proof-of-concept material that can be translated to vapor-phase gate and porous hosts. These provides the possibility of chemical sensors with memory, opto-chemical logic, non-volatile states with thermally gated erase. Finally, *Nature Communications* is a multidisciplinary forum bridging chemistry, condensed-matter physics, and functional materials. Given that ferroelectricity was not previously established in this [Fe₂Co] PBA system, and that our work significantly advances understanding of molecular ferroelectrics, we believe this platform will maximize constructive feedback from the ferroelectric research community and support the development of next-generation molecular ferroelectric devices.

Based on this point, we maintain that this work advances molecular ferroelectric science beyond our previous reports and deserves consideration at the present journal. We hope the reviewer will reconsider the broader conceptual contribution and multidisciplinary relevance of this study. We believe the results presented here represent a meaningful advance beyond our previous work and warrant publication in *Nature Communications*.

Reviewer #3 (Remarks to the Author):

Su, Wu, Sato and coworkers explore a trinuclear FeCo Prussian blue analogue which shows On/Off switching of ferroelectricity. The current compound is a new EtOH solvate of a known compound (Ref. 30) and as such represents another example in the series of trinuclear systems explored by these authors that all show a polar/non-polar state accessible thermally and by light. Moreover, recent references beyond the authors' own work reveal that solvent effects and SC-SC transformations are not new in PBA chemistry: <https://doi.org/10.1021/jacsau.5c00681> (an example of solvent effects in a 1D PBA) and Chem. Sci., 2025, 16, 130-138; a recent example of SC-SC transformation in a ETCST system.

However, these systems simply show ETCST and do not have the additional ferroelectric component. Hence, to the best of my knowledge this is the first time that anyone has demonstrated multiple switching in a molecular PBA analogue with on/off switching of ferroelectricity. The authors also make every effort to characterise the different phases upon thermal, light and EtOH cycling, clearly demonstrating the level of novelty and, for the most part, the experimental completeness I would expect for Nat. Commun.

Despite this there are areas of improvement in the paper. The so-called desolvated system, is in fact a dihydrate only system. I understand the desire to keep the nomenclature simple, but this description of the compound as being desolvated is inaccurate. I suggest another term is used perhaps dihydrate for $[\text{Fe}(\text{Tp})(\text{CN})_3]_2[\text{Co}(\text{dpa})_2] \cdot 2\text{H}_2\text{O}$ ($[\text{Fe}_2\text{Co}]$) and alcohol-dihydrate for $[\text{Fe}(\text{Tp})(\text{CN})_3]_2[\text{Co}(\text{dpa})_2] \cdot 2\text{H}_2\text{O} \cdot 1.5\text{EtOH}$ ($[\text{Fe}_2\text{Co}] \cdot 1.5\text{EtOH}$).

Response: Thank you for this thoughtful suggestion. To avoid the ambiguity of the term “desolvated,” we now use the following explicit nomenclature throughout: complex **1** = $[\text{Fe}(\text{Tp})(\text{CN})_3]_2[\text{Co}(\text{dpa})_2] \cdot 2\text{H}_2\text{O} \cdot 1.5\text{EtOH}$ (alcohol-dihydrate); complex **1'** = $[\text{Fe}(\text{Tp})(\text{CN})_3]_2[\text{Co}(\text{dpa})_2] \cdot 2\text{H}_2\text{O}$ (dihydrate). We state on first use that loss of co-crystallized ethanol from **1** gives **1'** via a single-crystal-to-single-crystal transformation. All instances of “desolvated” have been replaced with “EtOH-removed”. We believe this resolves the terminology issue while keeping the text concise.

I note that neither of the samples have elemental analysis, why is this? This is essential to ensure there are no amorphous impurities in the samples.

Response: Thank you for your comment. Since most of our measurements were performed on single-crystal samples, we overlooked including elemental analysis data. We fully agree that elemental analysis is important to exclude amorphous impurities. We have now performed elemental analysis for the pristine complex **1** (Elemental analysis calcd (%) for $\text{C}_{47}\text{H}_{51}\text{N}_{24}\text{O}_{3.5}\text{B}_2\text{Fe}_2\text{Co}$: C 47.03, H 4.28, N 28.01; Found: C 47.11, H 4.31, N 28.11) and added the results to the revised manuscript.

The crystallographic data looks OK, but I am a little concerned about the use of Mo in a system that crystallizes in a polar space group. Even though the Flack parameter is decent for the Pn structure, this can happen when Mo is used. Why did the authors not use Cu, where Flack parameters are more reliable?

Response: Thank you for this insightful comment. While Cu K α can be advantageous for very light-atom structures (like organic crystals) due to its stronger intensity and longer wavelength, it also increases the absorption coefficients for the 3d transition metals markedly. The reliability of the Flack parameter is governed by the precision of Bijvoet differences and control of systematics rather than wavelength. Our crystal contains Fe and Co atoms, which already provide sufficient resonant scattering at Mo K α . The present Mo dataset has excellent Friedel coverage (0.894; fraction_full 0.996), high redundancy (122,617/27,004), and low merging R_{int} (0.0472). Using the Flack quotient method on 8,580 Friedel quotients gives a Flack parameter of 0.063(5), indicating a small inversion-twin fraction consistent with ferroelectric domain formation. Moreover, independent SHG and pyroelectric measurements further confirm the polar space group. Given these metrics, Mo K α provides a reliable absolute-structure determination for this composition, while Cu K α would substantially increase absorption and the risk of systematic error owing to its larger absorption coefficients. We hope these explanations satisfactorily address your concerns.

The structure of [Fe₂Co]·1.5EtOH lacks hydrogen atoms on the EtOH molecules. I did a quick refinement, and I was able to add these and get the structure to settle, so the authors should do this. Similar issues seem to be present for the 30 K structures, where H atoms are missing for some of the solvent molecules. These can be added and refine well (put Hs on the water first then add to the EtOH to get it to settle), so there's no need to omit them. In both structures there are some clearly disagreeable reflections that should be omitted.

Response: Thank you for your helpful suggestion. We have revised the single-crystal data and updated the datasets deposited in the CCDC system according to your suggestion.

The structure of the dihydrate (i.e. -EtOH) is good, but there's a piece of analysis that I feel is missing. The hysteresis drops from 22 K to 13 K, but we are not provided with a structural rationalisation of this. What changes occur beyond the change to a C2/c group? What interactions replace those mediated by the EtOH? Do the chains of trinuclear units get closer together?

Response: Thank you for this insightful comment. Upon removal of EtOH, analysis of the unit-cell parameters shows a contraction along the *a* and *c* axes, while the *b* axis remains nearly unchanged. Simultaneously, the hydrogen-bonding network changes from an H₂O···EtOH···NC framework in **1** to a water-mediated network in **1'** (EtOH H-bond donors/acceptors are absent). This indicates that the trinuclear units move closer to each other in the absence of ethanol molecules.

With these quantitative observations in hand, we offer a cautious interpretation. In ETCST lattices, the thermal hysteresis width reflects the overall cooperativity including elastic, dipolar, and host-guest interactions that couple local ET/spin events across the 3D lattice. The increased packing density and loss of EtOH-mediated links in **1'** plausibly reduce the effective interchain coupling pathway relative to **1**, which is consistent with the observed narrowing of hysteresis from 22 K to 13 K. However, because cooperativity depends on the full network (including anisotropic elastic constants and domain-wall energetics), we refrain from asserting a direct conclusion on this point.

The authors use 785 nm light to access the MS state and I wondered if there's any particular rationale for this? In their latest 2024 Angew Chem paper (Ref. 31) the authors show that the IPA solvate shows different MS states depending on the wavelength of light used. It seems like a missed opportunity not to explore the same wavelength dependence here.

Response: Thank you very much for your thoughtful comment and for recognizing our previous work. In contrast to the IPA solvate, the present EtOH co-crystal (**1**) is mechanically fragile during the photo-induced phase transition: photo-cycling and especially multi-wavelength protocols systematically induce cracking, preventing us from reliable electric measurements. Therefore, we only used the excitation of 785 nm that reliably triggers a complete photoconversion in the photomagnetic measurements.

On the other hand, the scopes of these two studies are also different. In the present study, our goals are to establish macroscopic ferroelectricity, to demonstrate guest-regulated on/off ferroelectricity at the bulk-single-crystal level, and to clarify symmetry-changing pathways. A comprehensive wavelength survey would be valuable, however, this direction lies beyond the primary scope of the present manuscript and would not significantly reinforce the core conclusions demonstrated here. We consider such wavelength-dependent studies on a more robust sample for the future work.

Finally, the polarization change is a key highlight of this work, but it's lacking context. The value is found to 140 nC/cm², but how does this compare with other molecular ferroelectric materials? I note that the IPA solvate (Ref. 30) seems to be a little higher than the EtOH solvate. I think it would be a good idea for the authors to provide a table in the ESI that compares this system with the others that have been reported.

Response: Thank you for this insightful suggestion. In this work, our primary goal is to provide further evidence that this system represents a new electronic ferroelectric, and the reported polarization value is obtained on a pellet sample made from polycrystalline sample. Therefore, the current value reflects a partial polarization change owing to the crystalline/domain cancelation rather than the fully saturated polarization. At this stage, a direct quantitative comparison with other molecular ferroelectrics would not be appropriate, as it would not represent the intrinsic maximum polarization of this system.

We fully agree that such a comparison would be valuable once saturated *P–E* loop measurements on the single crystals are obtained. We are actively working toward performing such measurements to determine the saturated polarization and will include a comparative analysis in future work.

We appreciate the reviewer's thoughtful insight and believe that the suggested comparison will greatly benefit the field once complete data become available.

Overall, if the authors can update and improve the work, I can see it being a good paper for Nat Commun.

Minor points

1. Line 73, the phrase ‘.based on our previous work,’ is unreferenced, please correct.

Response: Thank you for your comment. We have revised this sentence to: “In this study, we developed a compound exhibiting directional ETCST by incorporating distinct solvent molecules based on our previous work³⁰”.

2. Figure 1 is missing the x-axis.

Response: Thank you for your comment. We have revised it.

3. In Figure 2, the use of blue for C and grey for N, is very unusual and could lead to widespread misunderstanding as these are typically the other way round. Previous work by these authors has used grey for C and blue for N, I see no reason to change this.

Response: Thank you for your comment. We fully acknowledge the conventional colour scheme where carbon is depicted in grey and nitrogen in blue. In the present work, however, we intentionally adopted a different colour scheme to clearly distinguish these figures from those in our previous publications on related systems, thereby helping readers easily differentiate between separate studies.

While the colour choice is unconventional, we believe it does not affect the interpretation of the structures, as all atom types are clearly labelled in the figures. For clarity, we have also added a note in the caption to explicitly state the atom colour assignments.

We hope this explanation clarifies our intention and appreciate your understanding.

4. Figure 5 is not as clear as it could be, can the authors not plot the data as they did in their 2024 JACS paper?

Response: Thank you for your helpful comment. We have further improved Figure 5 in the revised manuscript to enhance the clarity of the plotted information while maintaining the format that best highlights the characteristics of the current data set.

5. In Figure 6 caption, it would be a good idea to put in the net polarization value of $2.45 \mu\text{C}/\text{cm}^2$.
 Response: Thank you for the helpful suggestion. We have now added the net polarization value of $2.45 \mu\text{C cm}^{-2}$ to the caption of Figure 6 in the revised manuscript: “Black arrow, net polarization change ($2.45 \mu\text{C cm}^{-2}$)”.

6. Line 365, the formula should $\text{Bu}_4\text{N}[\text{Fe}(\text{Tp})(\text{CN})_3]$, with the 3 next to the CN ligands.
 Response: Thank you for your comment. We have revised it to $\text{Bu}_4\text{N}[\text{Fe}(\text{Tp})(\text{CN})_3]$: “and a solution of $\text{Bu}_4\text{N}[\text{Fe}(\text{Tp})(\text{CN})_3]$ (112 mg, 0.2 mmol)”.

7. What are the CCDC nos. of the structures and what program was used to solve the structures, SHELXS or SHELXT?
 Response: Thank you for your comment. We have now added the CCDC deposition numbers for all reported crystal structures in the revised manuscript: “CCDC 2352581-2352583, 2352585, 2352586 and 2377474 containing the supplementary crystallographic data can be obtained free of charge from the Cambridge Crystallographic Data Center via www.ccdc.cam.ac.uk/data_request/cif.”
 The structures were solved using SHELXT.

8. Ref. 44 is incorrect and should J Am Chem Soc.
 Response: Thank you for pointing this out. We have carefully re-checked Ref. 44, and confirm that the cited work was indeed published in *Inorganic Chemistry*, not *J. Am. Chem. Soc.* Therefore, the reference is correct as listed.

9. Figure S2 is a little fuzzy, please correct.

Response: Thank you for your comment. We've improved the resolution of this figure.

10. There is substantial noise in the UV-Vis in Figure S3 around 800-900 nm, why is this?

Response: Thank you for your comment. The noise observed in the UV-Vis spectrum around 800–900 nm in Figure S3 originates from the instrument light-source switching region. In this wavelength range, our spectrometer switches between different light sources, which inevitably introduces signal fluctuations.

11. Figure S5, should be: thermogravimetric analysis.

Response: Thank you for your comment. We have now revised the caption of Figure S5: “Thermogravimetric analyses (TGA) of **1** and **1'** under air flow.”.

Reviewer #1 (Remarks to the Author):

In the 2nd-round revision, the authors have made the detailed responses to all the reviews' concerns, and I think the manuscript has been well improved. So, I recommend it for publication in the current form now.

Response: Thank you for your recommendation. We have revised the manuscript and supporting information to incorporate feedback from the other reviewers, and we hope these revisions satisfactorily address all of your concerns.

Reviewer #3 (Remarks to the Author):

The revision by Su, Wu, Sato and coworkers adequately addresses most of the comments but there are still a few points that I think need work. Provided that these changes are made I would be happy to see the work published in nature Communications.

Specific points

1. Spin crossover is sometimes spelt as spin-crossover. I don't mind which the authors use as long as they are consistent.

Response: Thank you for the suggestion. We have unified the terminology throughout the manuscript and now consistently use "spin crossover".

2. Lines 188-189, 'When the sample were directly placed at a temperature below 60 K from room temperature, the $\chi_m T$ would be maintained at ca. $4.05 \text{ cm}^3 \text{ mol}^{-1} \text{ K}$, indicating the presence of a quenching effect.' is a little confusing as it suggests 60 K below RT which would be 238 K. What the authors really mean is that quench cooling results in trapping of the MS* phase, provided that the sample is kept below 60 K.

Response: Thank you for pointing out the ambiguity in the original wording. The sentence has been revised to clarify the meaning: "When the sample were directly placed at a temperature below 60 K from HT phase, the $\chi_m T$ would be maintained at ca. $4.05 \text{ cm}^3 \text{ mol}^{-1} \text{ K}$, indicating the presence of a quenching effect."

3. Having conducted DSC measurements, why are these not presented in the SI?

Response: Thank you for the suggestion. Although DSC measurements were presented in the response letter, we chose not to include them in the Supporting Information because the data quality does not allow reliable interpretation beyond a qualitative confirmation of the transition temperatures. As explained to reviewer 1, endothermic/exothermic features were observed at ~174 K (heating) and ~147 K (cooling), consistent with the ETCST temperatures determined by magnetometry (~170 K and ~148 K). However, the cooling transition occurs very close to the lower temperature limit of our DSC apparatus, where unavoidable thermal leakage significantly suppresses the exothermic signal. As a result, the peak area is not sufficiently reproducible for extracting thermodynamic parameters (ΔH , ΔS), and presenting the data could invite over-interpretation. Given that magnetometry already establishes the transition temperatures and hysteresis robustly, we believe including these DSC data would not meaningfully strengthen our conclusions.

4. The suggestion that in Figure 2, the authors cannot use a sensible colouring system for the crystal structures because readers will mistake the work for previous papers is nonsensical. The aim of figures is to communicate information quickly and accurately. Using blue for C almost guarantees that readers will misunderstand what the figures are trying to show and I must insist that this change

is made.

Response: Thank you for this comment. The coloring system in Figure 2 has been modified according to the reviewer's suggestion. A conventional and chemically intuitive color scheme is now used, ensuring that the structural information is clearly and accurately conveyed.

Figure 3. Molecular structure and space group switching of 1. The LT phase was in the polar space group Pn , whereas the HT, and MS* phases were in the nonpolar space group $P2/n$. The MS* phase was obtained by placing the single-crystal sample directly at 30 K, and no ETCST behavior occurred. With a temperature increase, the MS* phase returned to the LT phase via relaxation. During the thermal-induced ETCST process from the HT to LT phases, the orientation of part of the EtOH molecules changed from disorder to order. The gray dashed lines represent hydrogen bonds. The faintly drawn part is the adjacent structure with a disordered EtOH molecule connected by hydrogen bonds. Ethanol molecules are indicated with an ellipse (type A, red; type B, blue). Orange, Fe^{III}; green, Fe^{II}; purple, Co^{II}; pink, Co^{III}; yellow, B; gray, C; red, O; blue, N; light gray, H.

Similarly, Figures 5 and 7 have been revised as follows:

Figure 5. Molecular structure, space group, and spin and valence states of the metal centers between 1 and 1' during dynamic vapor sorption. a. Schematic representation of the on/off switching of ferroelectricity via desorption/adsorption of the guest molecule. The HT phase of 1 was in the nonpolar space group $P2/n$, the LT phase of 1 was in the polar space group Pn , while the LT* and HT* phases of 1' were in the nonpolar space group $C2/c$. In the LT* phase, the two irons were in the average state of Fe^{II}_{Ls} and Fe^{III}_{Ls} . The transition between the LT* and HT* phases was a temperature-dependent reversible process. Orange, Fe^{III} ; green, Fe^{II} ; purple, Co^{II} ; pink, Co^{III} ; yellow, B; gray, C; red, O; blue, N; light gray, H. b. Temperature-dependent $\chi_m T$ product of 1 (blue) and 1' (red). Open circle, heating process; filled circle, cooling process.

Figure 7. Directional ETCST induced ferroelectric phase transition and polarization switching mechanism. a. Schematic representation of 1 in the ferroelectricity "on" state via directional ETCST. The LT phase was in the polar space group Pn , whereas the HT phase was in the nonpolar space group $P2/n$. During the phase transition, the compound underwent the directional ETCST. Orange, Fe^{III} ; green, Fe^{II} ; purple, Co^{II} ; pink, Co^{III} ; yellow, B; gray, C; red, O; blue, N. Red arrow, direction of electron transfer; yellow arrow, polarization direction within the crystal lattice. b. Calculated polarization change from the directional charge transfer and EtOH reorientation. Green arrow, contribution from EtOH reorientation ($0.43 \mu C/cm^2$); Red arrow, contribution from directional electron transfer ($3.22 \mu C/cm^2$); Black arrow, net polarization change ($2.45 \mu C/cm^2$).

Figures S4, S7, and S8 in the Supporting Information have also been revised.

Figure S4. Molecular structure, spin and valence states of metal centers, and space group switching of **1**. The LT phase was in the polar space group Pn , whereas the HT, MS and MS* phases were in the nonpolar space group $P2/n$. The MS* phase was obtained by placing the high temperature phase directly at the low temperature, and no ETCST behavior occurred. With a temperature increase, the MS* phase returned to the LT phase via relaxation. The MS phase was obtained by irradiating LT phase by 785 nm light. During the thermal-induced ETCST process from the HT to LT phases, the orientation of part of the EtOH molecules changed from disorder to order. The gray dashed lines represent hydrogen bonds. The electron transfer process is indicated with a curved arrow. The blurred part is the adjacent structure with a disordered EtOH molecule connected by hydrogen bonds. Orange, Fe^{III}; green, Fe^{II}; lavender, Co^{III}; plum, Co^{II}; yellow, B; gray, C; red, O; blue, N; light gray,

Figure S7. Molecular structure, space group, and spin and valence states of the metal centers between **1** and **1'** during dynamic vapor sorption. Top: The HT phase of **1** was in the nonpolar space group $P2/n$, and the LT* and HT* phases of $[\text{Fe}_2\text{Co}]$ were in the nonpolar space group $C2/c$. In the LT* phase, the two irons were in the average state of $\text{Fe}^{\text{II}}_{\text{LS}}$ and $\text{Fe}^{\text{III}}_{\text{LS}}$. The transition between the LT* and HT* phases was a temperature-dependent reversible process. Bottom: Schematic representation of $[\text{Fe}_2\text{Co}]$ in the ferroelectricity “off” state due to nondirectional ETCST. Both the LT* and HT* phases were in the nonpolar space group $C2/c$. Orange, Fe^{III} ; green, Fe^{II} ; lime, Fe; gold, Co^{II} ; plum, Co^{III} ; yellow, B; gray, C; red, O; blue, N; light gray, H.

Figure S8. Molecular stacking and space group of **1** at LT, MS* and HT phase. The **1** crystallizes in a polar space group Pn in the LT phase, while the **1** crystallizes in the nonpolar space group $P2/n$ in the HT phase and MS* phase. During thermo-induced ETCST process, the disordered part of ethanol molecules in the HT phase become ordered in the LT phase. Orange, Fe^{III} ; green, Fe^{II} ; lavender, Co^{II} ; plum, Co^{III} ; yellow, B; gray, C; red, O; blue, N.